# *Azospirillum brasilense* Can Impressively Improve Growth and Development of *Urochloa brizantha* under Irrigation

**Bruno Rafael de Almeida Moreira** [1], **Ronaldo da Silva Viana** [2,*], **Vinícius Lopes Favato** [2],
**Paulo Alexandre Monteiro de Figueiredo** [2], **Lucas Aparecido Manzani Lisboa** [2],
**Celso Tadao Miasaki** [2], **Anderson Chagas Magalhães** [2], **Sérgio Bispo Ramos** [2],
**Charlene Raquel de Almeida Viana** [1], **Vanessa Dias Rezende Trindade** [1] and **André May** [3]

[1]  Department of Phytosanitary, Rural Engineering and Soils, School of Engineering, São Paulo State
    University (Unesp), Ilha Solteira, São Paulo 17385-000, Brazil; bruno_rafael.m05@hotmail.com (B.R.d.A.M.);
    charleneraquel@hotmail.com (C.R.d.A.V.); vanessatrindade@gmail.com (V.D.R.T.)
[2]  Department of Plant Production, College of Agricultural and Technological Sciences, São Paulo State
    University (Unesp), Dracena, São Paulo 17900-000, Brazil; vinicius.lopes@hotmail.com (V.L.F.);
    paulo.figueiredo@unesp.br (P.A.M.d.F.); lucas.lisboa@unesp.br (L.A.M.L.);
    celso.t.miasaki@unesp.br (C.T.M.); ac.magalhaes@unesp.br (A.C.M.); sergio.bispo@unesp.br (S.B.R.)
[3]  Brazilian Agricultural Research Corporation (Embrapa), Jaguariúna, São Paulo 13820-000, Brazil;
    andre.may@embrapa.br
*   Correspondence: ronaldo.viana@unesp.br; Tel.: +55-18-3821-7476

**Abstract:** Development of strategies to ensure grazing systems are sustainably produced in harsh environments, while not fertilizing them conventionally, is challenging. Figuring out the extent to which dose of inoculation and period of watering can positively influence the establishment of an effective symbiosis between *U. brizantha* cv. Marandu and *Azospirillum brasilense* is the point of this research. The treatment consisted of mixing 1 kg seeds with the inoculant of the strains Ab-V5 and Ab-V6 at 5, 10, 20, and 40 mL kg$^{-1}$, 2 x 10$^8$ CFU mL$^{-1}$. The plants grew in pots watered 2, 4, 8, and 16 days after sowing over thirty-days, twice. The bioagent at 5–10 mL kg$^{-1}$ enabled the plants watered up to 4 days after sowing to peak the production of dry mass of shoots (28.50 g) and roots (12.55 g). The efficiency of the symbiosis goes down quickly with increasing dose and delay of watering. Hence, if the dose of inoculant is higher than 10 mL kg$^{-1}$, it cannot successfully act in plants watered at least 8 days after sowing anymore. In conclusion, *A. brasilense* can assist in *U. brizantha* cv. Marandu growth and healthy development unless a lack of water in the substrate and an overdose collectively deter its potential.

**Keywords:** foraging-crop; palisade-grass; plant growth-promoting bacterium; rhizobacterium

## 1. Introduction

Brazil is one of the world's largest producers, exporters, and consumers of commodities. Biofuels, cellulose, grains, and meat are the main outcomes of the country's agriculture, forestry, and livestock segments for trade. Brazil is extensively covered with commercial pastures. Most of the farmers have not yet enough expertise and hands-on skills to implement and manage intensive farmyards sustainably. *Urochloa brizantha* is one of the most affordable, most reliable, and most cost-effective sorts of foraging crops to compose grazing systems in tropical zones like the middle west, north, and northeast Brazil. This cluster accounts for the world's largest cattle herd [1–4].

Experts classify *U. brizantha* into the family *Poaceae*. This is a specie of C$_4$ grass. Phenotypically, it looks like a strictly compact shrub. Its leaf structure is long and narrow, and its root system is large

in a specific surface. These morphoanatomical features collectively ensure that it tillers and regrows vigorously, even if it is anchored in substrates under harsh microclimates. *U. brizantha* is replete of appreciable benefits for the steadily rising world population. Economically, it can produce meat, milk, and wool without any difficulty. Environmentally, it is one of the simplest and wisest biosystems to assist in mitigating the emission of greenhouse gases into the atmosphere, while promoting and preserving long-lasting storage of organic carbon in the soil. Socially, it can offer employment opportunities and improve the conditions of people living in rural zones, where accessibility to goods and services is difficult [5–7].

Productivity and quality of forage of *U. brizantha* are likely to vary drastically with soil fertility and availability of water in the substrate. Water, nitrogen (N), and phosphorus (P) are crucial inorganic substances for the growth and development of the plant. These elements are the greatest sources of energy into biochemical pathways of synthesis of nucleic acids, pigments, hormones, and antimicrobial biocompounds. Mineral fertilizers have a lot of benefits for intensive pastures. However, they can be expensive and detrimental to natural ecosystems if they are used carelessly and injudiciously, rather than rationally. Development and implementation of cost-effective, harmless strategies targeting the replacement of conventional fertilizing chemicals to ensure pastures are sustainably produced are necessary for both economic and environmental reasons. Plant growth-promoting bacteria could be an option to do this safely and efficiently [8–10].

Plant growth-promoting bacteria, diazotrophic bacteria, or simply rhizobacteria are part of the group of microorganisms coexisting with autotrophic living things in symbiosis. Species of rhizobacteria can be endophytic, if they colonize through the root tissues, or free-living, if they live in the rhizosphere freely. Irrespective of the category, rhizobacteria offer key benefits to plants. Biological fixation of $N_2$ from the atmosphere, solubilization of organic phosphates, and secretion of phytohormones are their most memorable functions. Other advantages include synthesis of heavy metal-complexing siderophores, photosynthetic and photoprotective pigments, and control of herbivory pests and phytopathogens. Rhizobacteria can associate with mycorrhizal fungi, thus, changing the root system. The plant can end up more efficiently uptaking nutrients from substantial depths of the soil as a consequence of this friendly bacterial–fungal relationship. Essentially, the symbiont powers the plant, while the plant mutually releases carbohydrates, proteins, lipids, vitamins, and many other organic exudates for the growth and development of the symbiont. The success of the plant–rhizobacteria relationship depends on how synergistic the components are [11–19].

Several genera of rhizobacteria that improve both the productivity and quality of major and minor crops exist. *Azospirillum* sp. is among them. Species of *Azospirillum* sp. is greatly versatile and can enable grasses, such as rice, sugarcane, sorghum, and rice, to grow and develop healthy [20–26]. Data on the benefits of symbiosis by *A. brasilense* on the growth and development of species of *Urochloa* sp. under tropical microenvironments are available from the literature. Inoculation of seeds of *Urochloa* sp. with solutions consisting of the strains Ab-V5 and Ab-V6 at 15 mL kg$^{-1}$, 2 × 10$^8$ colony-forming units (CFU) mL$^{-1}$, can impressively improve the accumulation of N in the biomass upon drying [27]. On the contrary, the dynamics of tillering in *U. brizantha* cv. Marandu does not change significantly with spraying the same strains at 500 mL ha$^{-1}$, 2 × 10$^8$ CFU mL$^{-1}$ [28]. Dose and method of inoculation, whether mixing and spraying, as well as the microclimate, are factors determining the success of applying *A. brasilense* to *Urochloa* sp. The performance of this specie of *Azospirillum sp.* is clearly not consistent, and this requires further investigation.

Therefore, figuring out the extent to which the dose of inoculation and the period of watering can positively influence the establishment of an effective symbiosis between *U. brizantha* cv. Marandu and *A. brasilense* is the point of this research.

## 2. Materials and Methods

### 2.1. Infrastructure

Major and Minor Crops Division, College of Agricultural and Technological Sciences, São Paulo State University (Unesp), Campus of Dracena, São Paulo, Brazil.

### 2.2. Plant Material and Rhizobacterial Agent

The plant material was the cultivar *U. brizantha* cv. Marandu. The strains of *A. brasilense*, CNPSo 2083 (Ab-V5) and CNPSo 2084 (Ab-V6), made up the inoculant. The microbial material precisely was from the Microbiological Collection of the Brazilian Agricultural Research Corporation (Embrapa), Jaguariúna, São Paulo, Brazil. Temperature and relative humidity of the air during the initial storage of seeds and inoculant in hermetic polyethylene containers in a laboratory to prevent them from being contaminated by surrounding agents, which might influence their integrity upon growing, were in the optimal ranges of 22.5 ± 2.5 °C and 55 ± 5%, respectively, for 4 h [27].

### 2.3. Experiment

#### 2.3.1. Planning

The experiment was in a completely randomized block, 5 × 4 factorial, corresponding to five doses of inoculant (0, 5, 10, 20, and 40 mL kg$^{-1}$) and four periods of watering (2, 4, 8, and 16 days after sowing). Each test comprised of five replicates.

#### 2.3.2. Setting-Up

Preparation of Substrate

The experimental facility was an arch-type greenhouse, 6-m sides, with a cover of transparent plastic film of 50% light transmittance. Before setting up the experiment, samples of soil at 0–0.2 m depth were collected for chemical characterization (Table 1). The substrate was sun-dried, sieved in stainless-steel wire cloth to 2.5 mm, then stirred in a rotating chamber at 50 rpm, clockwise, for 10 minutes. The material was amended with 6.15 g $CO(NH_2)$, 8.15 g $CaSO_4(H_2PO_4)_2$, and 3.75 g KCl, and transferred into 10-L pots randomly placed on a tabletop, 1.5 m high from the floor.

**Table 1.** Chemical properties of the soil for the experimentation.

| Property | Unit |
|---|---|
| pH | 4.5 |
| Organic matter | 4.5 mg dm$^{-3}$ |
| P | 6 mmol$_c$ dm$^{-3}$ |
| K | 5.5 mmol$_c$ dm$^{-3}$ |
| Ca | 10 mmol$_c$ dm$^{-3}$ |
| Mg | 4 mmol$_c$ dm$^{-3}$ |
| S-SO$_4^{-2}$ | 7 mmol$_c$ dm$^{-3}$ |
| Potential acidity, H + Al$^{3+}$ | 18 mmol$_c$ dm$^{-3}$ |
| Al$^{3+}$ | 1 mmol$_c$ dm$^{-3}$ |
| Exchangeable cations | 19.5 mmol$_c$ dm$^{-3}$ |
| Cation exchange capacity | 37.5 mmol$_c$ dm$^{-3}$ |
| Saturation of exchangeable cations | 51.5% |

Biological Treatment

The materials previously stored in an oxygen biochemical chamber were naturally dried before further procedure. The biological treatment consisted of mixing 1 kg of seeds with the inoculant containing $2 \times 10^8$ CFU mL$^{-1}$ in a sterile environment until homogenization was achieved [27].

Sowing and Watering

The sowing consisted of placing ten seeds for each pot at a 0.025-m depth. During the experiment, in duplicate, plants were watered with deionized water accordingly to periods proposed to simulate the effect of availability of water in the substrate on the success of the efficiency of symbiosis. Removal of weeds was performed daily.

*2.4. Technical Analysis*

The plants were assessed according to architectural and ultrastructural traits, adapting methods of Figueiredo et al. [29] and Gírio et al. [30].

2.4.1. Architectural Traits

Height: H, expressed in centimeters, measured the vertical size of the plant;
Number of leaves: $N_L$, expressed as unit per pot, were visually counted;
Number of tillers: $N_T$, expressed as unit per pot, were visually counted;
Diameter of tiller: $D_T$, expressed in millimeters, measured the diameter of effective tiller;
Dry mass of shoots: the determination of $D_{MS}$, expressed in grams, consisted of drying up samples of leaves and tillers in a horizontal airflow drying-oven at 62.5 ± 2.5 °C, for 24 h, cooling down it, then weighing it to calculate the ratio of final and initial mass;
Dry mass of roots: the determination of $R_{DM}$, expressed in grams, followed the same method of determining $D_{MS}$.

2.4.2. Ultrastructural Traits

To assess the thickening of leaf epidermis ($T_E$), the thickening of leaf mesophyll ($T_{LM}$), the diameter of bulliniform epidermal cells ($D_{BEC}$), the diameter of bundle sheath cells ($D_{BSC}$), the diameter of xylem ($D_X$), and the diameter of phloem ($D_P$), a sample of leaves was excised into sections of 0.025 m × 0.025 m, then, immersed into a solution consisting of 37% formaldehyde, 70% acetic acid, and 70% ethanol for 24 h. The material was dehydrated, diaphanized, and placed onto histological slides, then sealed with albumin and stained with a solution of safranin at 1%. The ultrastructural traits were measured computationally in the environment of *CellSens Standards*. The software was calibrated to visualize high-definition microphotographs through an electronic microscopic embedded into the computer [31].

*2.5. Data Analysis*

The analysis of the data set formally started with running the procedures of Shapiro–Wilk and Bartlett to check if it was normal in distribution and homogeneous in variance. The one-way analysis of variance to test the effect of the dose of inoculant and period of watering on the architecture and ultrastructure of the plant material was performed. The Pearson product–moment correlation test to figure out potential linear relationships between variables was performed. Other methods of applying nontraditional mathematics to fit the data included 2D contour plotting. Before running it, we implemented fuzzy logic to turn eventual ambiguities off and improve the prediction and visualization of patterns defying understanding with classic Boolean logic. The software was *R-project* [32], which couples and runs on several platforms. This multiparadigm programming language provides a user-friendly environment for statistical computing and graphics.

## 3. Results

### 3.1. Effects of Dose of Azospirillum sp. and Period of Watering on the Architecture and Ultrastructure of Palisade-Grass

The sources of variation were not interactive to each other, regardless of the trait (Table 2). The architectural traits varied indirectly with both the dose of inoculant and period of watering. Most of the ultrastructural traits varied directly instead of indirectly.

**Table 2.** Analysis of variance for the effect of dose of *A. brasilense* and period of watering on the architectural and ultrastructural traits of *U. brizantha* cv. Marandu.

| Trait | Source of variation | | | Assumption | | CV; % |
|---|---|---|---|---|---|---|
| | Dose, A | Period, B | A × B | Shapiro–Wilk | Bartlett | |
| | *F*-value | | | *p*-value | | |
| Height | 25.25 * | 16.80 * | 0.50 | 0.10 * | 0.20 * | 4.65 |
| Number of leaves | 30.25 * | 39.60 * | 0.60 | 0.75 * | 0.10 * | 15.05 |
| Number of tillers | 105.90 ** | 71.85 * | 2.30 | 0.65 * | 0.05 * | 7.45 |
| Diameter of tiller | 50.20 * | 10.70 * | 0.05 | 0.30 * | 0.05 * | 12.65 |
| Dry mass of shoots | 305.65 ** | 78.55 * | 0.05 | 0.35 * | 0.15 * | 13.65 |
| Root dry mass | 15.20 * | 106.80 ** | 1.20 | 0.20 * | 0.30 * | 14.60 |
| Thickening of epidermis | 20.65 * | 12.85 * | 1.50 | 0.55 * | 0.15 * | 5.80 |
| Thickening of leaf mesophyll | 13.20 * | 5.50 * | 0.05 | 0.05 * | 0.05 * | 8.65 |
| Diameter of bulliniform epidermal cells | 5.65 * | 8.55 * | 0.05 | 0.05 * | 0.70 * | 13.65 |
| Diameter of bundle sheath cells | 75.20 ** | 17.80 * | 0.20 | 0.25 * | 0.40 * | 10.60 |
| Diameter of xylem | 20.65 * | 12.85 * | 1.75 | 0.45 * | 0.10 * | 15.80 |
| Diameter of phloem | 5.55 * | 28.90 * | 0.75 | 0.10 * | 0.05 * | 7.10 |

Significant code: * $p < 0.01$; ** $p < 0.05$. Coefficient of variation, CV.

The bioagent at 5–10 mL kg$^{-1}$ legibly enabled the plants anchored in pots watered up to 4 days after sowing to peak in height, production of leaves, tillers, and, obviously, dry mass of shoots and roots (Figure 1). On the contrary, doses and periods in the ranges of 10–40 mL kg$^{-1}$ and 8–16 days after sowing, respectively, caused the plants to be thicker in the epidermis and mesophyll of the leaf, and larger in bulliniform cells, bundle sheet cells, xylem, and phloem as well (Figure 2).

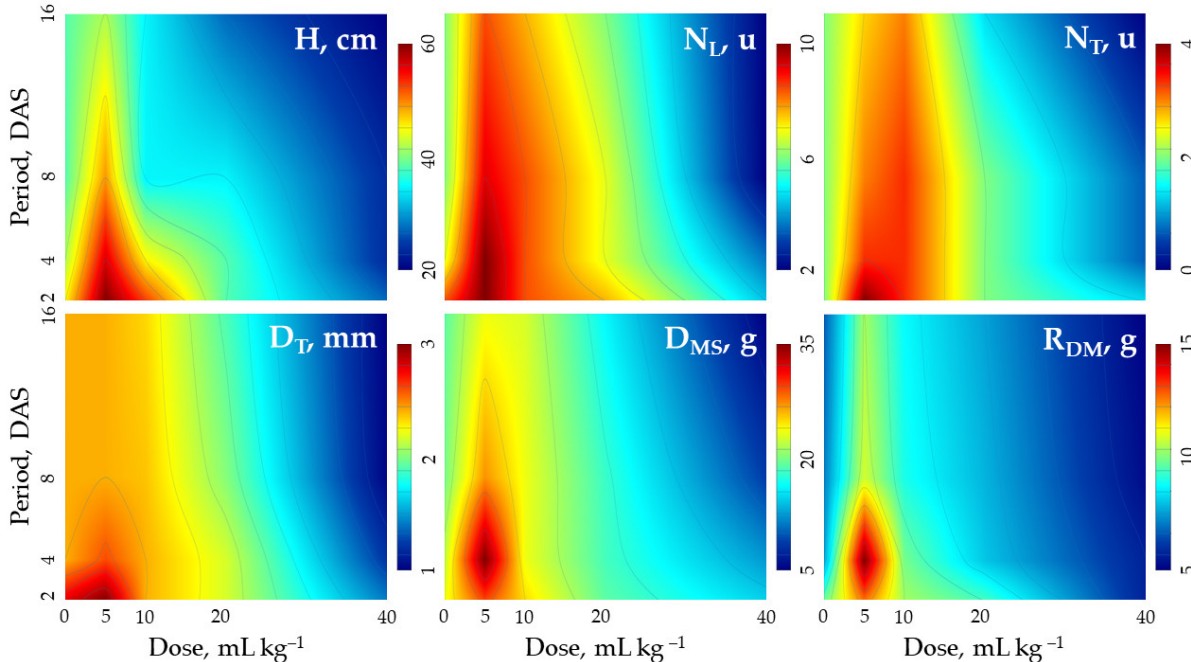

**Figure 1.** Detail-rich fuzzy colored card for the effect of doses of *A. brasilense* and period of watering on the architectural traits of *U. brizantha* cv. Marandu. Days after sowing, DAS; height, H; number of leaves, $N_L$; number of tillers, $N_T$; diameter of tillers, $D_T$; dry mass of shoots, $D_{MS}$; root dry mass, $R_{DM}$.

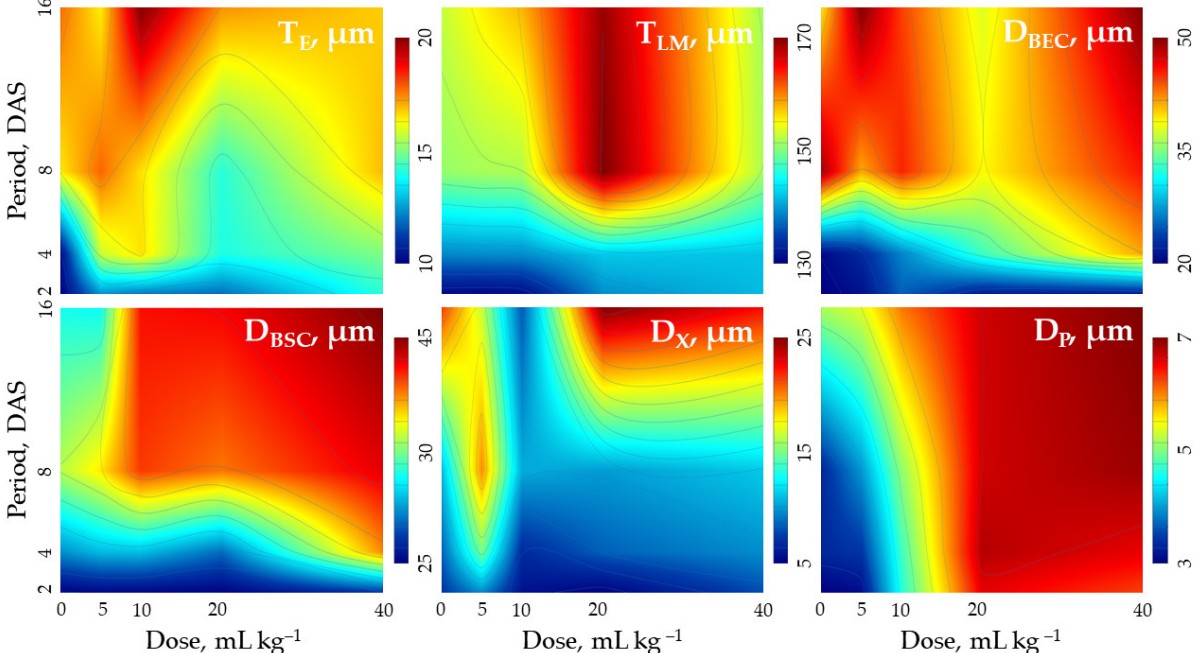

**Figure 2.** Detail-rich fuzzy colored card for the effect of dose of *A. brasilense* and period of watering on the ultrastructural traits of *U. brizantha* cv. Marandu. Days after sowing, DAS; thickening of epidermis, $T_E$; thickening of leaf mesophyll, $T_{LM}$; diameter of bulliniform epidermal cells; $D_{BEC}$; diameter of bundle sheath cells; $D_{BSC}$; diameter of xylem, $D_X$; diameter of phloem, $D_P$.

Practically, as long as the dose of inoculant is no higher than 10 mL kg$^{-1}$ and the period from sowing to watering is no larger than 4 days, the probability of *A. brasilense* successfully performing its functions and substantially improving the growth and development of *U. brizantha* cv. Marandu

is large. If the specie of foraging-crop is grown under longer unavailability of water in the substrate, it itself cannot grow and develop healthily anymore, unless the endophytic symbiont causes it to undergo substantial alterations in its vascular system and leaf morphoanatomy to an eventual adaptation to harsher microclimates.

### 3.2. Insights into the Growth and Development of U. brizantha cv. Marandu with A. brasilense under Watering

The Pearson product–moment test did not fail to track robustly the most salient (multi)collinear patterns from the data set (Figure 3). The H had positive correlations with $N_L$ ($r = 0.75$), $N_T$ ($r = 0.75$), $D_T$ ($r = 0.80$), $D_{MS}$ ($r = 0.85$), and RDM ($r = 0.85$), but a negative one with $D_{BEC}$ ($r = -0.75$). Therefore, the more robust the plant in size, the larger the probability of it producing and architecturally supporting more leaves and larger tillers unless it is channeling a larger amount of energy to develop larger bulliform cells to control stress by atypical transpiration stream, rather than in making photosynthetic and nonphotosynthetic parts above ground. The $D_T$ had positive linear relationships with both the $D_{MS}$ ($r = 0.90$) and $R_{DM}$ ($r = 0.90$). Plants higher in the mass of roots should, therefore, end up more strongly tillering and greatly accumulating photoassimilates in the biomass upon drying. The $T_{LM}$ and $D_{BSC}$ correlated positively ($r = 0.50$) to each other. Therefore, the thicker the mesophyll, the larger the environment of photosynthesis. Concretely, these multicollinearities endorse the trends of this study for the architectural and ultrastructural behavior of *U. brizantha* cv. Marandu growing with *A. brasilense* in symbiosis under periods of watering.

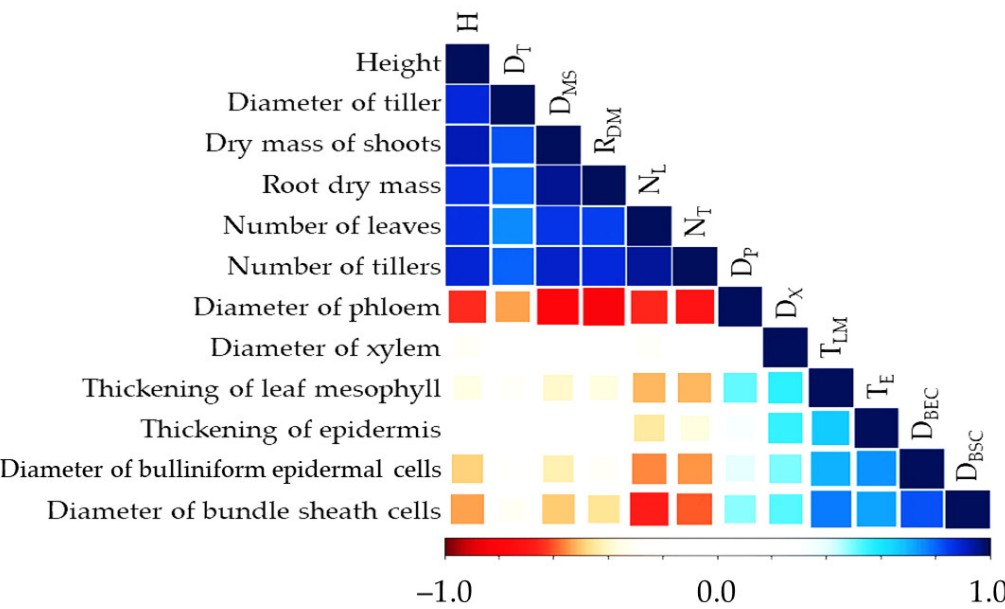

**Figure 3.** Correlogram for the linear relationships between architectural and ultrastructural traits of *U. brizantha* cv. Marandu growing with *A. brasilense* in symbiosis under periods of watering.

### 4. Discussion

The height of the plant is of great importance when screening potential species of forage to compose high-performance grazing systems. The depth of eating and digestive behavior by animals, as well as the size and success of producing either meat, milk, or wool, all depend on this architectural trait. Pasturelands higher in size usually provide high-quality feedstock, make eating by ruminants and nonruminants easier, and preventing the animals from contracting parasites and from spending lot of vital energy during the gathering of mass of forage near the soil [33–35]. The endophytic symbiont at 5–10 mL kg$^{-1}$ impressively enabled the plants to peak in height in substrate watered up to 4 days after sowing. An expressive gain in height would enable pastures of *U. brizantha* cv.

Marandu to end up more efficiently yielding high-quality masses of forage in tropical zones, unless a lack of water in the substrate deters its potential to be symbiotic with *A. brasilense*. The success of using A. *brasilense* at lower doses to increase the height in plants watered as early as possible is likely to be the result of it under microclimatically milder environments, more efficiently and consistently fixing $N_2$ from the atmosphere, solubilizing phosphates, or secreting phytohormones [36,37]. Age and physiological status of the plant, physicochemical properties of the substrate, and population density of cells are factors determining symbiosis. The denser the population of cells, the larger the probability of it intraspecifically competing for niches and energy resources from the environment. Hence, the plant–bacteria relationship cannot successfully perform symbiosis anymore [27]. This reference endorsed the trend of this study for the nonlinear effect of increasing doses of inoculant on the primary growth of *U. brizantha* cv. Marandu.

Species of foraging-crop higher in leaf to tiller ratios are often more nutritive, as they gain protein in biomass rather than in fiber. Additionally, they can develop themselves into grazing systems of high longevity and are mechanically resilient to the destructive forces of weather, high stocking rates, and subsequent cuts [12]. The use of *A. brasilense* at 5–10 mL $kg^{-1}$ caused the largest production of leaves in plants watered up to 4 days after sowing. Thereby, potential of this symbiont at lower doses and shorter periods of watering to casually assist *U. brizantha* cv. Marandu more efficiently withstand the adversities of heat, frost, animal trampling, and traffic of machineries on apical meristems, while ensuring some vigor to the regrowth, exists. The availability of N in the substrate is one of the most relevant abiotic factors affecting the production of leaves and tillers [3].

The tillering in grasses depends on the interactions between genotype, environment and management. Alterations in the morphoanatomy of root systems and biological fixation of $N_2$ by *A. brasilense* substantially improved both the tillering and density of tillers in species of *Urochloa sp.*, as pointed out by Castagnara et al. [38]. This citation supported noticeable enhancement in production of tillers in plants growing with *A. brasilense* at 5–10 mL $L^{-1}$ in pots watered up to 4 days after sowing. This finding is appreciable when dealing with the development and implementation of cost-effective strategies to optimize productivity and quality of feedstock, and mitigate potential losses of organic matter and minerals by hydraulic erosion in high-input pasturelands, as the percentage of cover varies directly with density of tillers.

The dry mass of shoots is the measurement of the mass of leaves and tillers upon drying. This variable is part of set of multiple morphophysiological criteria to assist precisely in choosing the sorts of foraging-crops to implement and manage pasturelands on a commercial scale. Species of foraging-crop greatly productive in the dry mass of shoots often ensure reliability of forage to feed large-scale herds at high stocking rates [39]. Inoculation of seeds with *A. brasilense* reflected higher accumulation of dry mass in shoots of corn and wheat [15,40]. These references supported the noticeable increment in the accumulation of dry mass in aboveground parts of *U. brizantha* cv. Marandu growing with this symbiont at 5–10 mL $kg^{-1}$, under periods from sowing to water of no larger than 4 days. The dry mass of shoots in plants in symbiosis by species of A. brasilense is likely to positively vary with the biological fixation of $N_2$ from the atmosphere, as the environment gains in the availability of readily assimilable forms of N, like $N\text{-}NO_3^-$ and $N\text{-}NH_4^+$. These ions power the accumulation of minerals in leaves and tillers upon drying [41]. Thereby, improvement in the production of dry mass of shoots would be another strength of growing *U. brizantha* cv. Marandu with *A. brasilense* towards the nutritive aspect of this summer grass.

The root dry mass is the measurement of the amount of mass in the root system upon drying. This is the simplest and most reliable indicator of effective symbiosis [42,43]. The use of *A. brasilense* in sugarcane significantly increased the accumulation of dry mass in root tissues, as pointed out by Chaves et al. [36] and Gírio et al. [30]. These reports were in line with the findings of this study for the marked increment in the production of dry mass of roots of *U. brizantha* cv. Marandu from seeds with inoculations of *A. brasilense* at 5–10 mL $kg^{-1}$, undergoing periods of watering in the range of 2–4 days after sowing. Integration of 5 mL $kg^{-1}$ and 2 days after sowing configured the best condition of dose of inoculant and period of watering to grow this specie of palisade-grass optimally. Additionally, acceptable technical performance of *U. brizantha* cv. Marandu with *A. brasilense* at the

lowest dose under periods of watering, in the range of 8–18 days after sowing, exists. This is of great importance when dealing with development and implementation of strategies to assist farmers creating, recreating, and fatting herds in drylands, where seasonality of rainfall makes the planning of livestock frameworks difficult, which declines both the productivity and quality of feedstock.

Water stress by drought disabled *U. brizantha* and *U. decumbens* from growing and developing healthily due to oxidation of vital metabolic pathways and physiological processes [44,45]. These references endorsed the adversities of longer periods running from sowing to watering on the technical performance of *U. brizantha* cv. Marandu from seeds with no inoculation. Practically, 8 and 16 days after sowing were the harshest periods of watering for the plant–soil–bacteria–atmosphere system. Yet, plants with *A. brasilense* at the lowest dose grew acceptably with a lack of water in the substrate, probably due to alterations in the features of morphoanatomy of the roots and leaves. Any substantial gain in the thickening of epidermis [45] and mesophyll [46–48], as well as in the diameter of bulliniform [49], bundle sheath [50–53] cells, xylem, and phloem [54], can lead to an improvement in the capacity of the plant to capture energy resources from the environment, and then storage them and convert them into biomass through the path of photosynthesis, even in low-water-content substrate. Evidence for the benefits of *A. brasilense* in the environment of photosynthesis and translocation of photoassimilates from photosynthetic to nonphotosynthetic parts of *U. brizantha* cv. Marandu exists. Further research tasks to better understand the extent to which the dose of inoculant and period of watering can change physiological processes in this specie of foraging-crop are, therefore, necessary.

## 5. Conclusions

The biological treatment of seeds by inoculation of *A. brasilense* at 5–10 mL kg$^{-1}$ can impressively improve both the growth and development of *U. brizantha* cv. Marandu unless the period from sowing to watering is longer than 8 days. If the dose of inoculant is higher than 10 mL kg$^{-1}$ and the period from sowing to watering is longer than 8 days, the plant–soil–bacteria–atmosphere system cannot successfully perform anymore as the functioning and technical efficiency of the endophytic symbiont is likely to decrease quickly with the concentrated population of cells in the rhizobacterial solution and the lack of water in the substrate. Yet, as long as the bioagent is able to promote substantial changes in the xylem, phloem, mesophyll, bundle sheath cells, and other features of vascular system and leaf morphoanatomy, the probability of *U. brizantha* cv. Marandu ending up more efficiently supporting eventual stresses is large. The findings of this timely research are of great importance when dealing with the development and implementation of cost-effective, harmless strategies to replace conventional fertilizer and ensure grazing systems are sustainably produced in harsh environments, like drylands, where the irregularity of rainfall makes the planning of intensive pasturelands difficult, which can decline the productivity and quality of feedstock for the production of meat, milk, and wool. To better understanding the extent to which dose of inoculant and period of watering can collectively alter the environment of photosynthesis, translocation of minerals, and photoassimilates, as well as the efficiency of the use of water in *U. brizantha* cv. Marandu for optimization of the concept, advanced analysis of morphometry, physiological processes, and biochemical reactions shall be the focuses of further research tasks.

**Author Contributions**: Conceptualization, R.S.V.; data curation, B.R.A.M. and C.T.M.; formal analysis, B.R.A.M. and C.T.M.; investigation, V.L.F., L.A.M.L., A.C.M., S.B.R., C.R.A.V., and V.D.R.T.; methodology, P.A.M.F., L.A.M.L., S.B.R., C.R.A.V., and A.M.; project administration, R.S.V. and P.A.M.F.; supervision, A.M.; writing— original draft, B.R.A.M.; writing—review and editing, R.S.V., A.C.M., S.B.R., and A.M. All authors have read and agreed to the published version of the manuscript.

**Funding:** This research received no external funding.

**Conflicts of Interest:** The authors declare no conflict of interest.

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
