# Peer review of "Azospirillum brasilense Can Impressively Improve Growth and Development of Urochloa brizantha under Irrigation"

_agriculture, doi:10.3390/agriculture10060220_

Round 1

Reviewer 1 Report

This manuscript describes  a study of  Azospirillum sp. to alternatively promote growth and development in palisade-grass. The results presented are only a number of plant architectural traits (also assessed ultrasonically), there is no data to investigate methods action, Azospirilllum plant growth promoting phenotype etc.

In my view this manuscript is incomplete to be considered for publication and only represents a subset of the data that would be required.

The manuscript shows only Table 3, are Tables 1 and 2 missing. Likewise, the only figure presented in Fig 5, where are the other 4?

The level of English requires considerable improvement. 

Reviewer 2 Report

The manuscript by Bruno Rafael de Almeida Moreira et al. describes the effect of seed inoculation of the palisade grass, Urochloa brizantha, with various doses of Azospirillum sp. on performance of the grass in the field in various drought regimes.

The manuscript is novel and describes some very interesting findings. Experimentation appears rigorous and interpretation and analysis are generally appropriate. However, I believe there are a number of issues with the manuscript which need to be addressed.

The bacterial inoculation is described only as Azospirillum sp.. This is too vague, greatly reducing the value of the findings. It gives little information to anyone wanting to make use of this finding or anyone wanting to replicate the experiment. Specific detail of the inoculum is required. If it is from a proprietary commercial mixture, at least the supplier details are required to ensure it is available to all.

Related to the above point, only one species of palisade grass is used so this specific species should be referred to in the text rather than the more general “Urochloa sp.”. At several points in the introduction or discussion Urochloa sp. is referred to when a specific species is intended. Please use the specific name. Similarly, the abstract should mention this species rather than referring to all palisade grasses.

Much of the key data is presented only in supplementary figures. This should be presented in figures in the main body of the manuscript. However, there are also some issues with these supplementary figures which need addressing. Firstly, their quality is not sufficiently high. The labels are too small to read as the authors mention in the legend. Perhaps a graphic editing package can be used to add larger labels to the existing figures. Secondly, the key does not seem to be consistent. S3 and S4 seem to have the red and green direction swapped versus S1 and S2. Thirdly, as this is discrete data, would 3D bar charts not be a better way to represent the data? The surface charts involve extrapolation and smoothing of the data meaning up to 0 days delay in watering. Particularly in the 3D plots, it is the 0 days point which is represented most clearly by these extrapolated charts, obscuring some of the data for other watering regimes. As a result, for many traits, including height, it looks as if they actually continue to increase with Azospirillum dose up to the maximum while the text indicates that they actually fall off again above 0.1mL L-1 so it is hard to verify the interpretation given in the text. Main manuscript figures ideally consisting of 3D bar charts showing the actual measurements are needed and would be much clearer. The surface response charts could then remain as supplementary.

There is also some over-interpretation of results. The abstract states, “The higher the dose of rhizobacterial solution, the larger the probability of cells intraspecifically competing for root niches and energy resources from the plant-soil-bacteria-atmosphere system”. This conclusion is repeated on lines 170 and 365. This was not measured. No measure at all was made of the colonization of root niches by the bacterium so it needs to be made clear that this is purely speculation. Similarly, Line 211 refers to “positive effect of changes in the architecture and functioning of root system”. This conclusion is repeated on line 378. However, this is also speculation. No analysis of the architecture and functioning of the root system was examined here. Just the total root dry mass. Again, Line 329 states “The thickening of leaf mesophyll by the endophytic symbiont certainly was another factor assisting plants under periods of watering larger than 4 days after sowing”. There is no way to be certain as no controlled experiment was carried out to test this theory.

The discussion is also very long with a lot of unnecessary basic plant physiology. There is no need to explain the make-up of the plant vasculature, for example.

Finally, the grammar throughout the manuscript needs careful attention.

Minor points

Line 26: Watering regime is described as “with watering 2, 4, 8, and 16 days after sowing”. This is ambiguous. It would be clearer to describe this as a “delay” before watering.

Line 27: This should be 0.05–0.1 mL L-1, not 0.05–1.0 mL L-1.

Line 59 What is “divine rusticity”?

Line 107: As above the should be described as a delay before watering

Line 140/143: “Initial” not “inicial”

Line 146 and others: “Bulliform” not “bulliniforme”

Line 195: “dry mass of shoots (r = 0.90)” not “(r = -0.90)”

Table 3: All abbreviations should be explained in the legend

Reviewer 3 Report

General comments.

The topic is relevant. In the introduction, there is not enough information about the successful use of biologics for plant growth. The methods of the experiment are not clear enough and raise questions. Statistical processing of results is decent. However, in my opinion, the Discussion is written too verbose and popular. It is necessary to give more comparisons with other studies on Azospirillum as biologics – such works have been previously described. It should be emphasized, what is the advantage of this particular study?

After reading the paper I have some specific comments given below. In my opinion, The authors need to extensively revise their manuscript before resubmission. (Major revision)

Specific comments

Lines 78-79  “…synthesis of photosynthetic and photoprotective pigments, such as lutein, neoxanthins, violaxanthins and zeaxanthins…” Rhizobacteria can synthesize photosynthetic (chlorophyll?) and photoprotective (carotenoid) pigments? Authors must provide the source of this information.

Line 101 Specify the source and storage location of the Azospirillum sp strain used.? The manuscript does not give enough information about the strain. Where was it taken from? What are its properties? If it is a commercial bioagent, provide its characteristics and information about manufacturer.

Line 107 What was the microbial cell titer (the number of microbial cells or CFU) of Azospirillum sp. in “rhizobacterial solution”? This is the most important question. At what microbial cell titer the positive effect was observed, and at what titer the effect was already negative?

Line 118 “commercial inoculant” - What commercial inoculant was used?

Lines 119-121 How long and why were the inoculated seeds stored? Why weren't they used immediately?

Lines 122-123 “The plants were watered according to the periods proposed to simulate effect of drought tress” (stress?) Please provide the references or details about “the periods proposed” and watering regime.

Figures and tables. The numbering of tables and figures presented in the manuscript should be corrected, including those presented in Supplemented materials: in the manuscript -Table 1, 2...., Fig. 1, 2...., in Supplemented materials - Table S1, S2....; Fig. S1, S2....

The list of references contains too many sources in Portuguese: 26 out of 58. This makes it difficult to assess the validity of citations.

Round 2

Reviewer 1 Report

The authors have still only presented a number of plant architectural traits  there is no data to investigate methods action, Azospirilllum plant growth promoting phenotype etc.

In my view this manuscript is incomplete to be considered for publication and only represents a subset of the data that would be required.

The authors have stated in their reply to reviewers that the data presented here is only a part of "analysis of morphometry, physiological processes, biochemical reactions and changes in rhizosphere of U. brizantha cv. Marandu growing with A. brasilense in symbiosis".

In my view these data are also required to generate manuscript for publication

Reviewer 2 Report

The revised manuscript is considerably improved versus the original submission.

The authors have addressed all of my points well.

I only have a few minor points which now need addressing:

In Section 2.5. Data analysis, the details of the R-packages used should be included. R-project is a software environment not the actual software itself. It is important to describe the specific software packages used for all steps in the analysis in sufficient detail to allow the analysis to be replicated by others.

Line 194: If H shows a negative correlation with DBEC, should the correlation not be (r = -0.75)?

Line 225: The reference should be numeric.

Line 278: A. brasilense should be italic

The English is also greatly improved but still needs some minor correction.

Reviewer 3 Report

The revised version of the manuscript by Moreira and co-workers has been reviewed. The authors have satisfactorily addressed all the comments and the manuscript has considerably improved. I have only minor comments on this manuscript:

Lines 68-70: «Other advantages include synthesis of heavy metal-complexing siderophores, photosynthetic and photoprotective pigments, and control of herbivory pests and phytopathogens.». I did not receive an answer to the question: Rhizobacteria can synthesize photosynthetic (chlorophyll?) and photoprotective (carotenoid) pigments? Authors must provide the source of this information.

Lines 224-225: «Hence, the plant-bacteria relationship cannot successfully perform symbiosis anymore [Nunes].» What does this link mean in square brackets?

Regarding the list of references. If the policy of an English-language journal allows the use of almost half of the references in Portuguese, I do not mind.

I believe that the manuscript can be accepted for publication.
